# Using the Theory of Planned Behavior to determine COVID-19 vaccination intentions and behavior among international and domestic college students in the United States

Cheng-Ching Liu[1]☺*, Jiying Ling[1]☺, Nagwan R. Zahry[2]‡, Charles Liu[3]‡, Ravichandran Ammigan[4]‡, Loveleen Kaur[1,5]

1 College of Nursing, Michigan State University, East Lansing, MI, United States of America, 2 Department of Communication, University of Tennessee at Chattanooga, Chattanooga, Tennessee, United States of America, 3 University Advising, Michigan State University, East Lansing, MI, United States of America, 4 College of Education & Human Development, University of Delaware, Newark, DE, United States of America, 5 BSN Student, College of Nursing, Michigan State University, East Lansing, MI, United States of America

☺ These authors contributed equally to this work.
‡ NRZ, CL and RA also contributed equally to this work.
* chengliu@msu.edu

**Data Availability Statement:** All relevant data are within the paper and its Supporting Information files.

## Abstract

Vaccination is the most effective strategy for preventing infectious diseases such as COVID-19. College students are important targets for COVID-19 vaccines given this population's lower intentions to be vaccinated; however, limited research has focused on international college students' vaccination status. This study explored how psychosocial factors from the Theory of Planned Behavior (TPB; attitudes, perceived behavioral control, subjective norms, and behavioral intentions) related to students' receipt of the full course of COVID-19 vaccines and their plans to receive a booster. Students were recruited via Amazon mTurk and the Office of the Registrar at a U.S. state university. We used binary logistic regression to examine associations between students' psychosocial factors and full COVID-19 vaccination status. Hierarchical multiple regression was employed to evaluate relationships between these factors and students' intentions to receive a booster. The majority of students in our sample (81% of international students and 55% of domestic students) received the complete vaccination series. Attitudes were significantly associated with all students' full vaccination status, while perceived behavioral control was significantly associated with domestic students' status. Students' intentions to receive COVID-19 vaccines were significantly correlated with their intentions to receive a booster, with international students scoring higher on booster intentions. Among the combined college student population, attitudes, intentions to receive COVID-19 vaccines, and subjective norms were significantly related to students' intentions to receive a booster. Findings support the TPB's potential utility in evidence-based interventions to enhance college students' COVID-19 vaccination rates. Implications for stakeholders and future research directions are discussed.

**Funding:** The authors would like to thank the Sherwood foundation (RN031103—LIUSF) who provided research funding to help us focus on the vaccination status among international and domestic students in the United States. The funders had no role in study design, data collection and analysis, decision to publish, or preparation of the manuscript.

**Competing interests:** The authors have declared that no competing interests exist.

## Introduction

The World Health Organization declared the coronavirus (COVID-19) outbreak a pandemic in March 2020. Almost seven million people worldwide and over one million Americans had lost their lives to the disease as of April 19, 2023 [1]. Vaccination remains the most effective means of mitigating and preventing infectious diseases such as COVID-19. Approximately 71% of the United States (U.S.) population aged 5 years and above are fully vaccinated against COVID-19, whereas less than 50% of the U.S. population aged 12 years and above have received their first booster [2]. COVID-19 vaccines help protect people from severe illness, hospitalization, and even death [3]. These vaccines also offer people who have had COVID-19 additional protection against hospitalization from a new infection. Receiving these vaccines can further contribute to community health via widespread immunization: vaccinated individuals indirectly protect the greater community by reducing person-to-person transmission. For example, once college students were sufficiently vaccinated against COVID-19, campuses began to resume in-person classes, large-group activities, sport programs, international exchanges, and social gatherings [4].

Many studies have been conducted on U.S. domestic college students' acceptance, reluctance, risk perceptions, and attitudes regarding COVID-19 vaccines. The findings have been inconsistent [4–6]. For instance, Kecojevic et al. noted that about 45% of surveyed domestic college students found information about COVID-19 vaccines hard to understand, but 60% agreed that they were responsible for getting vaccinated and protecting others from the disease [4]. Between 6% and 8% of college students surveyed concurred that young adults do not need to receive COVID-19 vaccines thanks to natural immunity. On the contrary, researchers observed that 47.5% of sampled college students ($n = 134$) who had not yet been vaccinated were hesitant to receive COVID-19 vaccines [5]. In another study, researchers found that 50% of unvaccinated college students did not intend to receive these vaccines due to lack of trust in the vaccines' efficacy, fear of side effects, and being suspicious of the U.S. government's intentions behind the large push for vaccination [6].

Approximately 1.1 million international students enrolled at U.S. colleges in 2021, and in the 2022/23 academic year, these institutions expect to receive even more international student applications [7]. However, these students constitute an under-researched and vulnerable population, specifically in the health domain [8]. Only a handful of studies in the U.S. have examined international students' vaccination status; of these, efforts have generally focused on human papillomavirus (HPV) vaccines. No research appears to have addressed international college students' COVID-19 vaccination status in the U.S. Immunization schedules and vaccine availability differ nationally, potentially affecting international students at U.S. higher education institutions. Many of these students also lack access to preventive vaccines in their home countries [9]. According to the Our World in Data website (October 9, 2022), approximately 70% of the global population has received at least one COVID-19 vaccine versus only 23% of residents in low-income countries [10]. As an example, merely 17.5% of people in Nigeria are fully vaccinated against COVID-19 [10]. Although 90% of the Chinese population has been fully vaccinated with China's CoronaVac and Sinopharm vaccines, scholars have questioned these immunizations' length of protection given their poor efficacy in preventing symptomatic disease [10, 11]. Moore et al. indicated that international students enrolling at United Kingdom (U.K.) universities may need to be updated on domestic immunization policies due to differing vaccination schedules [12]. Specifically, the odds of U.K. nationals being vaccinated were about five times higher than for international (non-U.K.) students attending U.K. universities [12]. This discrepancy could reflect variation in immunization schedules between the U.K. and international students' home countries. Moreover, researchers have

determined that vaccine hesitancy is higher in some minority communities, such as the international student population [4]. These circumstances collectively call for further research. Current inconsistencies especially highlight the need to clarify college students' attitudes towards and intentions to receive COVID-19 vaccines and boosters.

Barriers to vaccine hesitancy/refusal must be removed before effective health interventions and education about COVID-19 vaccines and boosters can be developed. Meanwhile, to increase immunization rates, it is essential to understand college students' vaccination status and intentions to receive COVID-19 vaccines and boosters. Factors associated with COVID-19 vaccination also stand to be uncovered. This study examined whether psychosocial factors drawn from the Theory of Planned Behavior (TPB), namely attitudes, perceived behavioral control (PBC), subjective norms, and behavioral intentions were related to full COVID-19 vaccination and the intentions to receive a booster. We compared responses among domestic and international college students in the U.S. Students are considered fully vaccinated if at least 2 weeks have passed since receipt of either the first Johnson & Johnson shot or the second Pfizer or Moderna shot [13].

## Theoretical framework

The TPB is a robust model that is predictive of human actions in specific contexts [14]. We adopted this theory to investigate whether attitudes, PBC, subjective norms, and behavioral intentions were associated with full COVID-19 vaccination and the intentions to receive a booster among international and domestic college students in the U.S. Attitudes refer to one's perceptions of a behavior as either favorable or unfavorable. PBC captures the extent to which a person believes they can execute a behavior. Subjective norms reflect social pressure to perform a behavior [14]. According to TPB, individuals' attitudes, PBC, and subjective norms can influence their intentions to engage in a behavior, which can lead to performing the behavior [14]. Researchers have applied the TPB to predict individuals' intentions and engagement in receiving vaccination against influenza, HPV, and COVID-19 [15–17]. For example, a recent meta-analysis of 43 articles found that TPB constructs including attitudes, PBC, and subjective norms predicted intentions to receive a COVID-19 vaccine among adult general population, parents, and patients [17]. Based on TPB, in our study, individuals' positive attitudes (e.g., "I believe that the COVID-19 vaccine[s] is effective"), PBC (e.g., "I have the time to get the COVID-19 vaccine[s]"), and subjective norms (e.g., "My family thinks I should get the COVID-19 vaccine[s]") were hypothesized to be associated with greater intentions to engage in COVID-19 vaccination. Moreover, greater intentions were assumed to be related to receiving a COVID-19 vaccine.

## Methods

### Research design and participant recruitment

This study was reviewed and classified as exempt by the hosting university's Institutional Review Board. This study involved a nonexperimental, cross-sectional design. The primary sample comprised college students between 18 and 26 years old. Two groups (i.e., domestic and international students) were nonrandomly selected to complete an online survey. Domestic students enrolled at U.S. higher education institutions were surveyed via Amazon Mechanical Turk (mTurk). International students were recruited from a large state university in the midwestern U.S. The university's Registrar Office sent out an email to all international students to complete a Qualtrics survey. In the email containing a link to the questionnaire, recipients were informed that the study concerned COVID-19 vaccines and boosters. Students were not required to provide their written informed consent to participate in this study;

however, students were required to read the consent information and check a box in the Qualtrics survey to certify that they agreed to participate in the study. Without checking the box, students would not be able to see and answer any questions. Students were also told that they could exit the survey at any time. All data was obtained anonymously.

Using the G*Power program, we assumed a small effect size with an odds ratio (OR) of 1.68 [18], 50% probability of not being fully vaccinated [2] and a significance level of 0.05, a sample size of 133 was needed to achieve at least an 80% study power when using logistic regression models to examine factors related to the full COVID-19 vaccination status. With a predicted 15% missing data rate, we over sampled to obtain at least 157 participants in each group. This sample size was also adequate for achieving a minimum of 80% study power when using hierarchical linear regression models to examine 17 variables that might be associated with intentions to receive a COVID-19 booster.

## Measures

Study measures were adapted from the TPB-based instrument developed by Catalano and colleagues [16]. The original version was developed to assess attitudes, PBC, subjective norms, and behavioral intentions in the context of HPV vaccination among college students [16]. The original instrument demonstrated acceptable readability for college students, good face and content validity via an expert panel review, acceptable test-retest reliability, and great internal consistency reliability with Cronbach's alphas ranging from 0.92 to 0.97 [16]. In addition, the original instrument had good construct validity through factor loadings of four constructs and acceptable predictive validity explaining 58% of the variance in intentions to receive HPV vaccine [16]. We adapted this instrument by replacing 'HPV vaccine' with 'COVID-19 vaccine,' while maintaining the same question items, to measure attitudes, PBC, subjective norms, and behavioral intentions in relation to receiving COVID-19 vaccines and boosters.

Respondents' demographic information (e.g., age, sex, ethnicity, education, health status, smoking history, other vaccination status) was also collected. For health status, participants were asked to answer if they were currently diagnosed with any diseases including diabetes, hypertension, autoimmune disease, hepatitis asthma, and cardiac disease; had a chronic medical condition requiring medication or regular physician visits; or had overall very good health with no disease. For smoking status, participants were asked to identify if they smoked, never smoked, or used to smoke but quit. Participants were also asked to identify if they had received one dose of the COVID-19 vaccine, they were fully vaccinated including one dose of Johnson & Johnson or two doses of Pfizer/Moderna, they had already received a booster shot, or they never received any COVID-19 vaccines. For general vaccination, participants were asked to indicate how up-to-date they were on vaccinations, with responses including: currently up-to-date, no but planning to be updated, or no and not planning to be updated.

Attitudes towards behavior receiving all COVID-19 vaccines and boosters were defined as a respondent's view of doing so as either favorable or unfavorable. Attitudes towards receiving all COVID-19 vaccines and boosters in the next 12 months were measured using seven items based on bipolar adjectives and scored on a 7-point Likert-type scale (e.g., "I think getting the COVID-19 vaccines and booster in the next 12 months would be. . .”; 1 = *not at all protective*, 7 = *extremely protective*). There are 7 questions in this subscale and the possible sum scores ranged from 7 to 49 (See Table 2).

PBC referred to the extent to which a respondent believed they could control receiving all COVID-19 vaccines and a booster in the next 12 months. This subscale included six items based on bipolar adjectives and scored on a 7-point Likert-type scale (e.g., "For me to get all COVID-19 vaccines and booster in the next 12 months would be. . .”; 1 = *extremely difficult*, 7

= *extremely easy*). There are 6 questions in this subscale and the possible sum scores ranged from 6 to 42.

Subjective norms were operationalized as one's general belief that significant people in their lives (e.g., family members or close friends) thought they should or should not get COVID-19 vaccines and a booster in the next 12 months. This subscale contained four items scored on a 7-point Likert-type scale (e.g., "My parents or legal guardians would like me to get the COVID-19 vaccines and booster in the next 12 months"; 1 = *completely disagree*, 7 = *completely agree*). There are 4 questions in this subscale and the possible sum scores ranged from 4 to 28.

Behavioral intentions reflected college students' intentions to receive all COVID-19 vaccines and a booster in the next 12 months. This subscale featured three items scored on a 7-point semantic differential scale (e.g., "I plan to get all COVID-19 vaccines and booster in the next 12 months"; 1 = *completely disagree*, 7 = *completely agree*). There are 3 questions in this subscale and the possible sum scores ranged from 3 to 21, with higher scores indicating greater willingness to receive COVID-19 vaccines and boosters.

## Data analysis

All analyses were performed in SPSS for Windows v28.0 (SPSS Inc., Chicago, IL, USA). Descriptive statistics returned counts and frequencies for categorical data and means and standard deviations for continuous data. Chi-square tests or independent *t* tests based on the focal variables' characteristics, were performed to determine variable-based differences between domestic and international students. Construct validity of the TBP-based instrument was assessed employing Principal Components with varimax rotation analysis. The numbers of factors were determined by Kaiser–Meyer–Olkin (KMO), scree plots, loadings of over .40, and explainable percentages of variance [19]. Cronbach's alpha was used to estimate the internal consistency reliability. Binary logistic regression was used to examine associations between TPB-based psychosocial factors (attitudes, PBC, subjective norms, behavioral intentions) and full COVID-19 vaccination status. Covariates such as respondents' age, sex, ethnicity/race, smoking behavior, chronic illnesses, health insurance, and vaccination history (including influenza) were also included in our model. All covariates were dummy coded except age. The four TPB factors were analyzed as continuous variables. Hierarchical multiple linear regression was conducted to evaluate the relationships of attitudes, PBC, subjective norms, and intentions to receive COVID-19 vaccines with intentions to receive a COVID-19 booster. For the hierarchical multiple linear regression model, demographic variables (i.e., age, sex, ethnicity/race, financial status, smoking behavior, chronic illnesses, health insurance, and vaccination history) were entered in Block 1; intentions to receive COVID-19 vaccines, attitudes towards receiving these vaccines, subjective norms, and PBC were entered in Block 2. Results were considered statistically significant at $p < 0.05$. Missing data were controlled by using group averages if participants missed answering 1 or 2 questions. If participants missed answering more than 3 questions, that participant was excluded from analysis.

## Results

### Participant characteristics

The sample included 197 international and 222 domestic college students. International students' mean age was 20.27 years (±2.75). About 44% were women (*n* = 86), 37% were freshmen (*n* = 73), 85% (*n* = 167) indicated they never smoked, 75% were up to date on general vaccines including influenza (*n* = 147), 70% were Asian (*n* = 137), 81% were fully vaccinated against COVID-19 (*n* = 160), and 4% (*n* = 8) had received a COVID-19 booster. Domestic students'

mean age was 24.62 years (±3.38); half were women ($n = 110$), 27% ($n = 59$) indicated that they smoked, 78% were up to date on other vaccines including influenza ($n = 172$), 77% were white ($n = 171$), 55% were fully vaccinated against COVID-19 ($n = 121$), and none had received a booster. As listed in Table 1, these student groups varied significantly in age, ethnicity, health

**Table 1. Demographics.**

| Variable | International Students ($n = 197$) | Domestic Students ($n = 222$) | Total ($N = 419$) | $p$ |
|---|---|---|---|---|
| Age | 20.27±2.75 | 24.62±3.38 | 22.53±3.91 | < .001* |
| Sex | | | | .386 |
| Male | 110 (55.8%) | 112 (50.5%) | 222 (53%) | |
| Female | 86 (43.7%) | 110 (49.5%) | 196 (47%) | |
| Ethnicity | | | | < .001* |
| Caucasian | 24 (12.2%) | 171 (77%) | 195 (46.5%) | |
| Asian | 137 (69.5%) | 27 (12.2%) | 164 (39%) | |
| Hispanic | 15 (7.6%) | 8 (3.6%) | 23 (5.4%) | |
| Black/African American | 10 (5.1%) | 7 (3.2%) | 27 (6.4%) | |
| American Indian | 0 | 7 (3.2%) | 7 (1.6%) | |
| Health status | | | | < .001* |
| Diabetes | 1 (.5%) | 26 (11.7%) | 27 (6.4%) | |
| Hypertension | 1 (.5%) | 36 (16.2%) | 37 (8.8%) | |
| Autoimmune disease | 2 (1%) | 19 (8.6%) | 21 (5%) | |
| Hepatitis | 0 | 6 (2.7%) | 6 (1.4%) | |
| Asthma | 3 (1.5%) | 16 (7.2%) | 19 (4.5%) | |
| Cardiac disease | 0 | 2 (1%) | 2 (.5%) | |
| Other chronic conditions | 3 (1.5%) | 3 (1.4%) | 6 (1.4%) | |
| Healthy overall | 75 (88.8%) | 109 (49.1%) | 284 (67.8%) | |
| Smoking status | | | | .010* |
| Yes | 15 (7.6%) | 59 (26.6%) | 74 (17.7%) | |
| No, never | 167 (84.8%) | 145 (65.3%) | 312 (74.5%) | |
| Past smoker | 8 (4.1%) | 13 (5.9%) | 21 (5%) | |
| Currently up to date on vaccinations | 147 (74.6%) | 172 (77.5%) | 319 (76%) | .783 |
| Have health insurance | 186 (94.4%) | 170 (76.6%) | | < .001* |
| School | | | | < .001* |
| Agriculture | 5 (2.5%) | 4 (1.8%) | 9 (2%) | |
| Arts & Letters | 0 | 22 (9.9%) | 22 (5%) | |
| Business | 36 (18.3%) | 30 (13.5%) | 66 (16%) | |
| Communication Arts | 15 (7.6%) | 22 (9.9%) | 37 (8.8%) | |
| Education | 6 (3%) | 30 (13.5%) | 36 (8.6%) | |
| Engineering | 62 (31.5%) | 67 (30.2%) | 129 (31%) | |
| Human Medicine | 2 (1%) | 11 (5%) | 13 (3%) | |
| Law | 2 (1%) | 2 (1%) | 4 (1%) | |
| Music | 2 (1%) | 2 (1%) | 4 (1%) | |
| Natural Science | 28 (14.2%) | 7 (3.2%) | 35 (8.4%) | |
| Nursing | 1 (.5%) | 1 (.5%) | 2 (.5%) | |
| Osteopathic Medicine | 2 (1%) | 2 (1%) | 4 (1%) | |
| Social Science | 23 (11.7%) | 14 (6.3%) | 37 (8.8%) | |
| Other | 9 (4.6%) | 6 (2.7%) | 15 (3.6%) | |
| College year | | | | < .001* |
| Freshman | 73 (37.1%) | 24 (10.8%) | 97 (23%) | |
| Sophomore | 33 (16.8%) | 32 (14.4%) | 65 (16%) | |
| Junior | 49 (24.9%) | 72 (32.4%) | 121 (29%) | |
| Senior | 38 (19.3%) | 92 (41.4%) | 130 (31%) | |
| Marital status | | | | < .001* |
| Married/living with partner | 5 (2.5%) | 90 (40.5%) | 95 (22.7%) | |
| Divorced/separated | 1 (.5%) | 0 | 1 (.2%) | |
| Single | 150 (76.1%) | 100 (49.1%) | 259 (61.8%) | |
| In a relationship | 38 (19.3%) | 23 (10.4%) | 61 (14.6%) | |

Note.

*Results were statistically significant.

status, smoking status, health insurance, marital status, and COVID-19 vaccination status.
Domestic students were older than international students. Most domestic students were white
($n$ = 171, 77%) whereas most international students were Asian ($n$ = 137, 69.5%). More inter-
national students ($n$ = 160, 81.2%) were fully vaccinated against COVID-19 compared with
domestic students ($n$ = 121, 54.5%). More domestic students ($n$ = 108, 49%) than international
students ($n$ = 9, 4.5%) reported having chronic illnesses. For instance, 36 domestic students
(16.6%) had been diagnosed with hypertension, 26 (12%) had diabetes, 19 (8.8%) had an auto-
immune disease, and 16 (7.4%) had asthma.

## TBP instrument reliability and validity

**Construct validity.**   Results from the factor analysis support the good construct validity of
the adapted instrument. The KMO was 0.91 indicating sampling adequacy [20]. Bartlett's test
of sphericity was statistically significant ($p$ < .001). Four factors were extracted and explained
69.35% of the total variance in Table 2.

**Internal consistency reliability.**   Cronbach's alpha was employed to assess internal consis-
tency in this study. The Cronbach's alpha for attitudes toward behavior subscale was .93
among international students and .89 among domestic students. For the PBC subscale, it was
.85 for international students and .86 for domestic students. The Cronbach's alpha for subjec-
tive norms subscale was .94 for international students and .82 for domestic students. For the
behavioral intentions subscale, Cronbach's alpha was .98 for international students and .92 for
domestic students.

## TBP-based psychosocial factors

As demonstrated in Table 3, international students scored higher on attitudes towards receiving
COVID-19 vaccines, subjective norms, and intentions to receive these vaccines but not on PBC.
Similarly, international students had more positive attitudes and stronger intentions to receive a
COVID-19 booster than domestic students. A statistically significant difference manifested
between international and domestic students' attitudes towards receiving COVID-19 vaccines
and boosters as well as intentions to receive these immunizations. No statistically significant dif-
ferences emerged between these student groups in terms of subjective norms or PBC.

## TPB-based factors and receiving full COVID-19 vaccines

Table 4 presents the results of binary logistic regression, taking full COVID-19 vaccination as a cat-
egorical variable (i.e., either fully vaccinated or not). Domestic students who were male (OR = .31,
95% CI: .16-.63; $p \leq$ .001) had lower odds of being fully vaccinated, consistent with our findings
when combining both student groups (OR = .60, 95% CI: .38-.95; $p$ = .029). Domestic students
who had higher PBC (OR = 1.10, 95% CI: 1.02–1.18; $p$ = .016) had higher odds of being fully vacci-
nated, similar as the results among all students (OR = 1.08, 95% CI: 1.04, 1.13; $p$ < .001). Interna-
tional students who had higher scores on attitudes had higher odds of being fully vaccinated
(OR = 1.07, 95% CI: 1.00–1.15; $p$ = .049); however, international students who had higher inten-
tions had lower odds of being fully vaccinated (OR = .88, 95% CI: .77–1.00; $p$ = .042). Among all
students, those who were older (OR = .87, 95% CI: .81-.94; $p \leq$ .001) and smoking (OR = .48, 95%
CI: .26-.89; $p$ = .019) demonstrated lower odds of being fully vaccinated against COVID-19.

## TPB-based factors and intentions to receive a COVID-19 booster

Results of the hierarchical multiple regression are summarized in Table 5. In the first step of
analysis (Block 1), demographics (i.e., ethnicity/race, sex, age, college year, smoking status,

**Table 2. TPB model items, factor loadings, and average variance extracted (AVE).**

| TPB constructs and corresponding items | Factor Loadings | AVE |
|---|---|---|
| **Attitudes toward behavior** | | 60.1% |
| 1. I think getting all doses of the COVID-19 vaccines in the next 12 months would be Very Bad ~ Very Good | 0.77 | |
| 2. I think getting all doses of the COVID-19 vaccines in the next 12 months would be Not at all protective ~ Extremely Pprotective | 0.76 | |
| 3. I think getting all doses of the COVID-19 vaccines in the next 12 months would be Unnecessary ~ Necessary | 0.75 | |
| 4. I think getting all doses of the COVID-19 vaccines in the next 12 months would be Very unhealthy ~ Very healthy | 0.77 | |
| 5. I think getting all doses of the COVID-19 vaccines in the next 12 months would be Disadvantageous ~ Advantageous | 0.86 | |
| 6. I think getting all doses of the COVID-19 vaccines in the next 12 months would be Painful ~ Painless | 0.69 | |
| 7. I think getting all doses of the COVID-19 vaccines in the next 12 months would be Extremely harmful ~ Extremely beneficial | 0.78 | |
| **Perceived Behavioral Control** | | 54.7% |
| 1. If I wanted to, I am sure I could get all doses of the COVID-19 vaccines in the next 12 months. Completely unsure ~ Completely sure | 0.76 | |
| 2. For me to get all doses of the COVID-19 vaccines in the next 12 months would be Extremely difficult ~ Extremely easy | 0.77 | |
| 3. How much control do you have to get all doses of the COVID-19 vaccine in the next 12 months? No control ~ Completely control | 0.73 | |
| 4. I am confident I can get all doses of the COVID-19 vaccines in the next 12 months, even if there is a financial cost. Very unconfident ~ Very confident | 0.72 | |
| 5. I am confident I can get all doses of the COVID-19 vaccines in the next 12 months, even if my schedule is busy. Very unconfident ~ Very confident | 0.69 | |
| 6. I am confident I can find a healthcare provider (for example, clinic, health center, physician's office) where I can get all doses of the COVID-19 vaccines in the next 12 months. Very unconfident ~ Very confident | 0.73 | |
| **Subjective Norms** | | 60.8% |
| 1. Most people who are important to me think that I should get all doses of the COVID-19 vaccines in the next 12 months. Completely disagree ~ Completely agree | 0.72 | |
| 2. My parent(s) or legal guardian(s) would like me to get all doses of the COVID-19 vaccines in the next 12 months. Completely disagree ~ Completely agree | 0.79 | |
| 3. Family members other than my parent(s) or legal guardian(s) (for example, sibling, aunt, uncle, grandparent, etc.) would like me to get all doses of the COVID-19 vaccines in the next 12 months. Completely disagree ~ Completely agree | 0.87 | |
| 4. My friends would like me to get all doses of the COVID-19 vaccines in the next 12 months. Completely disagree ~ Completely agree | 0.71 | |
| **Behavioral Intentions** | | 87.7% |
| 1. I intend to get all doses of the COVID-19 vaccines in the next 12 months. Completely disagree ~ Completely agree | 0.93 | |
| 2. I will try to get all doses of the COVID-19 vaccines in the next 12 months. Completely disagree ~ Completely agree | 0.93 | |
| 3. I plan to get all doses of the COVID-19 vaccines in the next 12 months—Completely disagree ~ Completely agree. | 0.93 | |

health insurance, chronic illnesses, financial status, and vaccination history) explained only 8.4% of the variance in all students' intentions to receive a COVID-19 booster. Having chronic illnesses (B = 1.26, $p$ = .038) increased students' intentions to do so whereas being male (B = -1.27, $p$ = .012) decreased it. In the second step (Block 2), demographics, attitudes, subjective norms, PBC, and intentions to receive COVID-19 vaccines explained 68% of the variance in

**Table 3. Comparisons between domestic and international students.**

| Variable | International Students Mean ± SD | Domestic Students Mean ± SD | $\chi^2/t$ statistic | $p$ |
|---|---|---|---|---|
| COVID-19 vaccine(s) | | | 77.644 | $< .001^*$ |
| Haven't received any | 6 (3%) | 13 (5.9%) | | |
| Received 1 dose | 23 (11.7%) | 88 (39.6%) | | |
| Fully vaccinated | 160 (81.2%) | 121 (54.5%) | | |
| Received booster | 8 (4.1%) | 0 | | |
| Intentions to receive COVID-19 vaccine | 16.54±5.26 | 14.27±5.02 | 20.232 | $< .001^*$ |
| Intentions to receive COVID-19 booster | 16.78±5.13 | 14.9±4.84 | 14.719 | $< .001^*$ |
| Attitudes towards COVID-19 vaccine | 42.11±7.61 | 39.27±7.39 | 14.999 | $< .001^*$ |
| Attitudes towards COVID-19 booster | 40.51±8.82 | 38.58±7.54 | 5.800 | $.016^*$ |
| Subjective norms | 22.24±5.65 | 21.77±4.26 | .931 | .335 |
| PBC | 32.09±6.58 | 32.62±5.86 | .773 | .380 |

Note.

*Results were statistically significant; PBC: perceived behavioral control. For definitions of COVID-19 vaccines, please refer to Measures.

intentions to receive a booster. Students' intentions to receive COVID-19 vaccines (B = .61, $p < .001$), attitudes towards receiving these vaccines (B = .07, $p = .004$), and subjective norms (B = .13, $p < .001$) were significantly related to their intentions to receive a booster.

Upon comparing both student groups, no demographic variables were significantly correlated with international students' intentions to receive a COVID-19 booster, while smoking (B = 1.64, $p = .037$) was significantly related to domestic students' intentions to do so. The model containing demographics, attitudes, subjective norms, PBC, and intentions to receive COVID-19 vaccines explained 65% of the variance in international students' intentions to receive a booster and 72% of that for domestic students. International students' intentions to receive COVID-19 vaccines (B = .53, $p < .001$) and subjective norms (B = .18, $p = .002$) were significantly related to their intentions to receive a booster. Only domestic students' intentions

**Table 4. TBP-based factors of fully receiving COVID-19 vaccines.**

| Factors | International Students | | | Domestic Students | | | All Students | | |
|---|---|---|---|---|---|---|---|---|---|
| | OR | 95% CI | $p$ | OR | 95% CI | $p$ | OR | 95% CI | $p$ |
| Age | .88 | .76, 1.03 | .102 | .96 | .86, 1.07 | .430 | .87 | .81, .94 | $< .001^*$ |
| Sex (male) | 1.70 | .78, 3.72 | .186 | .31 | .16, .63 | $< .001^*$ | .60 | .38, .95 | $.029^*$ |
| Ethnicity (Hispanic) | .55 | .12, 2.53 | .441 | 2.35 | .44, 12.43 | .315 | 1.13 | .40, 3.15 | .821 |
| Race (Asian) | .72 | .27, 1.92 | .514 | .32 | .08, 1.35 | .121 | 1.43 | .85, 2.42 | .182 |
| Chronic illness (yes) | 1.05 | .19, 5.69 | .956 | .87 | .44, 1.71 | .683 | .58 | .34, 1.01 | .056 |
| Updated vaccination (yes) | 1.68 | .70, 4.01 | .247 | 2.19 | .91, 5.28 | .079 | 1.73 | .99, 3.03 | .056 |
| Smoking (yes) | .56 | .16, 2.01 | .373 | .59 | .27, 1.29 | .185 | .48 | .26, .89 | $.019^*$ |
| Health insurance (yes) | 7.66 | .44, 132.81 | .162 | .50 | .21, 1.20 | .123 | 1.22 | .61, 2.44 | .583 |
| Attitudes | 1.07 | 1.00, 1.15 | $.049^*$ | 1.06 | .99, 1.13 | .121 | 1.08 | 1.04, 1.13 | $< .001^*$ |
| Behavioral intentions | .88 | .77, 1.0 | $.042^*$ | .97 | .90, 1.05 | .455 | .97 | .92, 1.03 | .326 |
| PBC | .99 | .92, 1.06 | .678 | 1.10 | 1.02, 1.18 | $.016^*$ | 1.02 | .98, 1.06 | .419 |
| Subjective norms | 1.03 | .93, 1.14 | .528 | .99 | .88, 1.10 | .814 | .96 | .90, 1.03 | .256 |

Note.

*Results were statistically significant; CI: confidence interval; PBC: perceived behavioral control. For definitions of chronic illness (health status), updated vaccination, and smoking status, please refer to Measures.

**Table 5. TBP-based factors of intentions to receive a COVID-19 booster.**

| Factors | International Students Block 1: R² = .08 | | Domestic Students Block 1: R² = .15 | | Combined Group Block 1: R² = .08 | |
|---|---|---|---|---|---|---|
| | B (95% CI) | p | B (95% CI) | p | B (95% CI) | p |
| Age (18–22) | 1.72 (-.87, 4.31) | .192 | .69 (-1.04, 2.42) | .432 | .27 (-.96, 1.51) | .662 |
| Sex (male) | -.84 (-2.32, .64) | .265 | -.64 (-2.14, .86) | .402 | -1.27 (-2.27, -.27) | **.012*** |
| Asian (Other) | .64 (-2.98, 4.26) | .728 | -1.38 (-5.18, 2.42) | .476 | .12 (-2.43, 2.68) | .924 |
| White (Other) | -.2.16 (-4.25, 3.82) | .916 | -.90 (-4.35, 2.53) | .603 | -.89 (-3.49, 1.70) | .498 |
| Black (Other) | -2.29 (-7.09, 2.51) | .347 | .68 (-4.09, 5.47) | .777 | -1.05 (-4.41, 2.30) | .538 |
| Hispanic (Other) | 2.81 (-1.56, 7.20) | .206 | -1.35 (-5.97, 3.27) | .565 | 1.02 (-2.14, 4.18) | .527 |
| Chronic illness (Healthy) | -.53 (-3.17, 2.10) | .690 | 1.34 (-.02, 2.70) | .054 | 1.26 (.07, 2.46) | **.038*** |
| Updated vaccination (Updated) | -.03 (-1.68, 1.61) | .967 | 1.65 (-.03, 3.34) | .055 | .87 (-.30, 2.05) | .147 |
| Smoking (Smoker) | -.66 (-3.38, 2.06) | .633 | 1.64 (.10, 3.18) | **.037*** | .67 (-.65, 1.99) | .318 |
| Received COVID-19 vaccine | -.89 (-2.56, .76) | .289 | 1.48 (.10, 2.85) | **.035*** | .85 (-.15, 1.86) | .097 |
| Financial status (Scholarship) | .62 (-.94, 2.19) | .434 | -1.41 (-2.97, .14) | .075 | -.004 (-1.06, 1.05) | .995 |
| Parent education (< college) | -1.11 (-2.69, .45) | .162 | .55 (-.92, 2.04) | .459 | -.18 (-1.24, .87) | .730 |
| Health insurance | 1.29 (-2.92, 5.51) | .545 | .55 (-1.03, 2.15) | **.049*** | 1.35 (-.09, 2.80) | .095 |
| | Block 2: R² = .65 | | Block 2: R² = .72 | | Block 2: R² = .68 | |
| Attitudes | .09 (.02, .16) | .141 | .05 (-.02, .13) | .153 | .07 (.02, .12) | **.004*** |
| Behavioral intentions | .53 (.43, .64) | **< .001*** | .64 (.56, .73) | **< .001*** | .61 (.55, .68) | **< .001*** |
| PBC | .05 (-.02, .13) | .163 | .05 (-.02, .13) | .203 | .04 (-.01, .10) | .071 |
| Subjective norms | .18 (.06, .29) | **.002*** | .10 (-.01, .23) | .096 | .13 (.05, .21) | **< .001*** |

Note.

*Results were statistically significant; CI: confidence interval; PBC: perceived behavioral control.

to receive COVID-19 vaccines (B = .64, $p < .001$) were significantly correlated with their intentions to receive a booster.

## Discussion

This study examined whether TPB-based factors (i.e., attitudes, PBC, subjective norms, and behavioral intentions) were related to college students' full COVID-19 vaccination status and intentions to receive a COVID-19 booster among domestic and international college students in the U.S. Even though not all TPB-based factors were statistically significantly related to college students' full COVID-19 vaccination status or intentions to receive a COVID-19 booster, the TPB-based factors together explained about 65% of the variance in intentions to receive a COVID-19 booster. Therefore, our results support the utility of TBP in explaining college students' COVID-19 vaccination intentions and behavior, and some variations were observed between domestic and international students.

Another potential explanation for the difference is that the higher rates of having chronic illnesses (e.g., hypertension, diabetes, autoimmune diseases, and asthma) among domestic students might have contributed to their vaccination hesitancy due to fear of side effects. This explanation is further supported by prior literature. For example, Vallée indicated that 28.7% of French people with HIV were hesitant to receive COVID-19 vaccines due to concerns about their overall health, chronic disease status, and vaccine-related side effects [21]. Rakusa discovered that people with multiple sclerosis and other autoimmune disorders were reluctant to be vaccinated against COVID-19 out of fear of side effects and potentially worsening neurological status [22]. However, based on guidelines from the Centers for Disease Control and Prevention (CDC), people with certain chronic illnesses (e.g., diabetes, cancer, heart disease, lung

disease, or immune system disease) should consider receiving COVID-19 vaccines because these immunizations can best protect them from serious illness or death from the virus [23]. More efforts are therefore needed to reduce misinformation about the relationship between chronic conditions and COVID-19 vaccines, especially among U.S. domestic students.

## COVID-19 vaccination differences between domestic and international students

Surprisingly, international students' COVID-19 vaccination rates were significantly higher than for domestic students. Government policies may have informed this difference: as of November 8, 2021, international students holding an F-1 or J-1 nonimmigrant student visa (i.e., not U.S. citizens) were required to show proof of full vaccination before flying to the U.S. [24, 25]. This standard did not apply to U.S. citizens, U.S. nationals, U.S. lawful permanent residents, immigrants, or (under certain circumstances) air crew members [24]. Most states also required colleges to accommodate domestic students exempted from vaccination for medical or religious reasons.

## TPB-based factors and receiving full COVID-19 vaccines

This study found that being male had lower odds of being fully vaccinated. COVID-19 is more likely to kill men than women in the U.S., yet many men in this country are not overly eager to be vaccinated [26, 27]. Slightly more than three-quarters (76.1%) of women aged 18 and above were fully vaccinated against COVID-19 compared with 71.5% of men between August 29 and October 30, 2021 [27]. Women may be more inclined than men to be proactive about public health issues and to pursue preventive health care. This outcome counters the results of a systematic review published in 2022 that analyzed data from 46 studies (141,550 participants): in 58% of cases, men had higher intentions than women to receive COVID-19 vaccines [28]. Zhong also pointed out that men held more positive attitudes and subjective norms than women regarding the importance of COVID-19 vaccination [29].

Unexpectedly, smokers had lower odds of being fully vaccinated, although research reported that smoking was associated with a higher risk of COVID-19 infection and with greater risks of all outcomes and hospitalization [30]. Similarly, patients who smoked more than 30 pack-years had 2.25 times higher odds of being hospitalized and were 1.89 times more likely to die after a COVID-19 diagnosis compared with people who had never smoked [31]. Although not statistically significant, our results indicate that college students with chronic illnesses had lower odds of being fully vaccinated compared to those without chronic illnesses. As we discussed earlier, individuals with chronic illnesses are hesitate of receiving COVID-19 vaccines due to the fear of side effects. Smokers and people with chronic diseases represent priority groups for COVID-19 vaccination, so programs focusing on reducing misinformation especially about the relationship between chronic illnesses and vaccines are needed to improve the vaccination rate.

Our study found that college students who were older had lower odds of being fully vaccinated which contrasts with two other studies that found higher vaccination rates among older adults compared with younger adults [32, 33]. However, our finding is consistent with a CDC report showing that college-aged young adults who were 25 or older were less likely than those aged 18 to 24 to get COVID-19 vaccines [34]. Additionally, a study investigated seasonal influenza vaccination coverage from four thousand college students in North Carolina found that being a freshman was related to the receipt of influenza vaccine because they were usually required to live on campus and were frequently receiving university announcements about the

availability of vaccines and other resources [35]. Therefore, college campus efforts are needed to focus on older college students to improve their accessibility to vaccines.

In this study, PBC was the sole TPB-based factor that was significantly related to vaccination against COVID-19 in domestic students. According to Ajzen, PBC is a joining concept that combined perceived control (i.e., the level of control an individual has over gotten vaccinated) with self-efficacy (i.e., people's confidence in their ability to vaccinate [36]). Our finding showed that college students with higher PBC had higher odds of being fully vaccinated which is consistent with some prior studies [36–39]. For example, Hayashi et al. reported that PBC was the most robust factor of TPB in their study that predicted American adults' intentions to take a COVID-19 vaccine [38]. Moreover, research showed that PBC was a very strong predictor to the intentions of people living with HIV to receive the COVID-19 vaccination [39]. Thus, programs targeting domestic college students' PBC may be very promising in improving their vaccine uptake.

Positive attitudes were significantly related to full vaccination against COVID-19 in international and all student groups. Researchers pointed out that individuals with more positive attitudes were more willing to receive vaccines [40]. However, a prior study showed that individuals with positive attitudes toward COVID-19 vaccines had lower acceptability of the vaccine due to concerns on long-term side effects, lack of transparent information about the vaccine, and vaccine hesitancy [41, 42]. In order to improve individual's attitudes toward COVID-19 vaccines, healthcare professions need to 1) understand individuals' health concerns; 2) provide reliable information related to side effects of the vaccines; and 3) foster strong partnerships with local health departments [42, 43].

Surprisingly, our study found that international students with higher intentions had lowed odds of being fully vaccinated. This unexpected result occurred may be because in our study, 81.2% of international students were already fully vaccinated and they did not need any intentions to receive the vaccine. In addition, the study data are cross-sectional, so results could not imply any causal relationships between interventions and vaccination behavior. Given that even individuals with higher intentions to vaccinate, they may not get vaccinated ultimately because of vaccine hesitancy [44]. Intentions to vaccinate is a critical determinant of COVID-vaccine uptake and can be influenced by personality traits, individuals' perception of the vaccine, individuals' trust in the government, individuals' perceived vulnerability to the disease, and their conscientiousness and demographic background [45–48]. It is important to study the moderators of the intention-behavior gap to increase vaccine uptake.

## TPB-based factors and intentions to receive a COVID-19 booster

For all students, individuals' attitudes, subjective norms, and intentions to receive COVID-19 vaccine were significantly associated with their intentions to receive a booster. The TPB framework maintains that PBC is vital to understanding behavior. However, in our study, it was not a significant factor related to intentions to receive a COVID-19 booster for either domestic or international students. One reason is the small percentages of students receiving a booster when the study was conducted: eight international and no domestic students received the booster.

Students' subjective norms (e.g., perceived social pressure) can shape their behavioral intentions and subsequent decisions about whether to engage in a behavior [30]. For instance, students' friends, family members, and colleagues could inspire them to get a vaccine. In our study, the subjective norms of COVID-19 vaccination were significantly related to students' intentions (i.e., among international students and the combined student population) to receive a COVID-19 booster. International students were more likely to say that most people they

knew and most of their family and friends would like them to receive COVID-19 vaccines and a booster. International students' subjective norms may have been also more pronounced than domestic students' due to the CDC vaccine mandate. Family members of international students and university offices for international students and scholars may encourage these students to receive COVID-19 vaccines to be able to enter the U.S. for school.

Intentions to receive COVID-19 vaccines were significantly correlated with the intentions to receive a COVID-19 booster among all students. International students showed greater intentions than domestic students to be vaccinated and boosted. The largest β value in the combined student population was tied to behavioral intentions (0.61), underscoring the importance of this construct. Even though immunization reduces the spread of COVID-19, its effectiveness depends on individuals' willingness to receive the vaccine [37, 48]. Webb and Sheeran stated in a meta-analysis that health-related intentions are causally associated with respective health-related behaviors [49]. Ajzen cited intentions as the most critical factor influencing actual behavior [36]. Unsurprisingly, if intentions are excessively low in either the general population or the student population, then the possibility of halting the COVID-19 pandemic drastically declines.

## Limitation

This study has several limitations. International students were recruited from a single higher education institution due to limited access during the pandemic, which constrains our findings' generalizability. Additionally, data were acquired in October 2021, when many colleges did not require a COVID-19 booster. Even though eight international students had received a booster, we did not have sufficient uptake data to analyze. Study data were also gathered via a self-report survey hosted on Qualtrics. Social desirability bias and students' personal beliefs may have influenced their responses. The cross-sectional nature of the study limits the inferences on causal relationships between TPB constructs and behavior. A more rigorous longitudinal study design is recommended to further evaluate the effects of TPB constructs on vaccination behavior and the relative contributions of each individual construct.

## Conclusions

Our study aimed to understand whether TPB-based psychosocial factors (i.e., attitudes, PBC, subjective norms, and behavioral intentions) were related to full COVID-19 vaccination status and the intentions to receive a booster among international and domestic college students in the U.S. PBC was a significant factor related to the full vaccination status among domestic college students, while attitudes were a significant factor associated with all students' full vaccination status. Moreover, attitudes towards receiving a COVID-19 booster, subjective norms, and intentions to receive COVID-19 vaccines were significantly correlated with the intentions to receive a booster. From a theoretical perspective, our findings somewhat support that the TPB is an applicable framework for explaining COVID-19 vaccination intentions and behavior among both domestic and international college students. Thus, public awareness and educational programs aimed at promoting vaccine acceptance should consider using TPB as a framework and tailor to each group. To enhance uptake of COVID-19 vaccines and boosters and to reduce the incidence of severe cases, healthcare providers and educators can develop vaccine campaigns to promote COVID-19 vaccination among college students. For example, universities and colleges can provide bilingual health interpreters, implement campus-based marketing strategies, send reminders using social media, and offer free and affordable vaccines to increase vaccination rates among international students [50]. When promoting the vaccination rate of domestic students, healthcare providers and educators need to understand

students' health concerns and answer their questions with evidence to increase their PBC [17]. We believe that the COVID-19 vaccine mandate has been crucial to increasing international students' COVID-19 vaccination rates. Higher education professionals and policymakers should thus reconsider college immunization requirements to improve vaccination rates.

## Supporting information

**S1 Data set.**
(XLS)

## Acknowledgments

We would like to thank the research participants who gave their valuable time of completing the study. Special thanks to Dr. Catalano and her research team for providing us permission for utilizing the instrument in our study.

## Author Contributions

**Conceptualization:** Cheng-Ching Liu, Jiying Ling, Nagwan R. Zahry.

**Data curation:** Cheng-Ching Liu, Jiying Ling, Nagwan R. Zahry.

**Formal analysis:** Cheng-Ching Liu, Jiying Ling, Loveleen Kaur.

**Funding acquisition:** Jiying Ling, Charles Liu, Ravichandran Ammigan.

**Investigation:** Cheng-Ching Liu, Charles Liu, Ravichandran Ammigan.

**Methodology:** Cheng-Ching Liu, Jiying Ling, Nagwan R. Zahry.

**Project administration:** Cheng-Ching Liu, Charles Liu, Ravichandran Ammigan.

**Software:** Cheng-Ching Liu, Jiying Ling, Loveleen Kaur.

**Supervision:** Cheng-Ching Liu, Jiying Ling, Loveleen Kaur.

**Validation:** Cheng-Ching Liu, Jiying Ling.

**Visualization:** Cheng-Ching Liu, Jiying Ling, Loveleen Kaur.

**Writing – original draft:** Cheng-Ching Liu, Jiying Ling, Loveleen Kaur.

**Writing – review & editing:** Cheng-Ching Liu, Charles Liu, Ravichandran Ammigan.

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
