## [Decision Letter · Decision Letter 0]

30 Jun 2023

PONE-D-23-12951Using Theory of Planned Behavior to Determine COVID-19 Vaccination Intention and Behavior among International and Domestic College Students in the United StatesPLOS ONE

Dear Dr. liu,

Thank you for submitting your manuscript to PLOS ONE. After careful consideration, we feel that it has merit but does not fully meet PLOS ONE’s publication criteria as it currently stands. Therefore, we invite you to submit a revised version of the manuscript that addresses the points raised during the review process.

We look forward to receiving your revised manuscript.

Kind regards,

Le An Pham, Ph.D,MD

Academic Editor

PLOS ONE

Journal Requirements:

"We would like to thank the Sherwood foundation (RN031103—LIUSF) who provided research funding to help us focus on the vaccination status among international and domestic students in the United States.."

"no"

4. Thank you for stating the following financial disclosure: "No"

5. Thank you for stating the following in your Competing Interests section: "no"

6. Please note that in order to use the direct billing option the corresponding author must be affiliated with the chosen institute. Please either amend your manuscript to change the affiliation or corresponding author, or email us at plosone@plos.org with a request to remove this option.

7. In your Data Availability statement, you have not specified where the minimal data set underlying the results described in your manuscript can be found. PLOS defines a study's minimal data set as the underlying data used to reach the conclusions drawn in the manuscript and any additional data required to replicate the reported study findings in their entirety. All PLOS journals require that the minimal data set be made fully available. For more information about our data policy, please see http://journals.plos.org/plosone/s/data-availability.

8. Please amend either the title on the online submission form (via Edit Submission) or the title in the manuscript so that they are identical.

**Additional Editor Comments:**

PBC abbreviate for what in abstract?

Please explain that why author use median of the four TPB factors for making the cutoff that divide respondents into low- and high-scoring groups.?

Please have clear definition of variables that combine some criteria such as chronic diseases, smoking, finance…and also table 2 intended vaccination COVID 19, intended to take booster (tables 2: Received 1 dose; Fully vaccinated; Received booster) that make confuse..

Please report the result of OR with 95% CI

Please explain clear about “Domestic students who were male (OR: 3.22, p = .001) and had lower PBC (OR: 8.33, p = .049) had lower odds of being fully vaccinated?”

Please explain that why author use determinant instead of risk for OR?

Please Explain that TBP model is appropriate to measure intended to take booster Vaccine COVID 19 when R =0.08?? Whether TBP model appropriate to all participants such as LMIC? They familiar with HBM than TBP

Reviewers' comments:

Reviewer's Responses to Questions

**Comments to the Author**

1. Is the manuscript technically sound, and do the data support the conclusions?

Reviewer #1: Yes

Reviewer #2: Partly

2. Has the statistical analysis been performed appropriately and rigorously? 

Reviewer #1: Yes

Reviewer #2: Yes

3. Have the authors made all data underlying the findings in their manuscript fully available?

Reviewer #1: Yes

Reviewer #2: No

4. Is the manuscript presented in an intelligible fashion and written in standard English?

Reviewer #1: No

Reviewer #2: Yes

5. Review Comments to the Author

Reviewer #1: 4. Conclusions are presented in an appropriate fashion and are supported by the data.

Conclusions are presented in a clear outline of 2 sections pertaining to the summary of the findings and implications for health interventions. However, there are still places for improvement. The conclusion only presented a part of the results through the wrap-up of associated factors with vaccination only among domestic college students and all students. Those findings relating to international students should be summarized and key differences between 2 groups should be highlighted. Based on this, the author might emphasize the need for health interventions that are exclusive for each group. Such practical implications will provide more actionable insights for policymakers, healthcare professionals, and educators.

5. The article is presented in an intelligible fashion and is written in standard English.

Although the study utilizes a robust methodology that yields important and relevant findings, the presentation of the discussion lacks clarity and coherence. The discussion elaborates the relationship between psychosocial elements of the TPB (i.e., attitude, behavioral intention, perceived behavioral control, and subjective norms) and full COVID-19 vaccination, as well as intentions to receive a booster, in both domestic and international students. However, the information pertaining to the associated factors with full vaccination or the booster is intertwined throughout the discussion, making it difficult to distinguish between them. Similarly, the information describing the determinants among domestic and international students is mixed within paragraphs, hindering the viewers' ability to discern the key differences. Seemingly, determinants of full Covid-19 vaccination among domestic students include PBC, chronic illnesses and smoking. Regarding the reception of boosters in domestic students, the intention to receive the COVID-19 vaccine predicts the intention to receive the booster. However, viewers might have difficulty identifying associated factors as these pieces of information were scattered throughout the discussion. Furthermore, there is repetition of certain points in discussion, further impeding its clarity and coherence. Several repeated points and their location are listed below:

Paragraph 1: International students COVID-19 vaccination rates were significantly higher than for domestic students. Later in paragraph 1: More international students (81.2%) than domestic students (55%) in our sample had already received COVID-19 vaccines

Paragraph 5: PBC significantly predicted domestic students’ full vaccination, reflecting confidence in their ability to be vaccinated against COVID-19. Paragraph 9: PBC was the sole TPB variable to significantly predict vaccination against COVID-19 among domestic students and the combined student sample in this study.

To enhance the clarity and cohesiveness of the discussion, I suggest a thorough rewrite in which a clear outline of TBP associated factors and primary determinants of full covid-19 vaccination as well as booster acceptance should be provided in 2 separate sections dedicating to international and domestic students. To improve the overall readability of the discussion, the author should avoid repetition and unnecessary redundancy.

Reviewer #2: The author should clarify the methodology of study especially an TPB-based instrument development.

This finding has helped health managers more information to develop health policies to prevent Covid-19 infection.

6. PLOS authors have the option to publish the peer review history of their article (what does this mean?). If published, this will include your full peer review and any attached files.

Reviewer #1: No

Reviewer #2: No

---

## [Author Response · Author response to Decision Letter 0]

18 Aug 2023

Dear reviews,

Thank you so much for all the points you have addressed. We have revised our manuscript to address all the points. We re-did the data analysis and Discussion to enhance clarity of the manuscript. We uploaded a document with fully explanations. Please see "Responses to the reviewers comments." Please let us know if there is anything we can do it better. 

Many thanks.

---

## [Decision Letter · Decision Letter 1]

6 Oct 2023

Using the Theory of Planned Behavior to determine COVID-19 vaccination intentions and behavior among international and domestic college students in the United States

PONE-D-23-12951R1

Dear Dr. liu,

We are pleased to inform you that your manuscript, "Using the Theory of Planned Behavior to determine COVID-19 vaccination intentions and behavior among international and domestic college students in the United States," has been judged scientifically suitable for publication and will be formally accepted for publication once it meets all outstanding technical requirements. Congratulations! 

In closing, thank you very much again for working with us to bring this work to fruition. We hope that as a member of the PLOS ONE community, you will continue to help the journal as a reviewer, use our articles (including your own) as a source of citation in your future work, and continue to submit your best work on PLOS ONE. In particular, please do try to cite the most recent of the published articles in PLOS ONE.

Congratulations again, and thank you for your contribution to PLOS ONE!

Kind regards,

Khalid Mehmood

Academic Editor

PLOS ONE

Additional Editor Comments (optional):

Reviewers' comments:

Reviewer's Responses to Questions

**Comments to the Author**

1. If the authors have adequately addressed your comments raised in a previous round of review and you feel that this manuscript is now acceptable for publication, you may indicate that here to bypass the “Comments to the Author” section, enter your conflict of interest statement in the “Confidential to Editor” section, and submit your "Accept" recommendation.

Reviewer #1: All comments have been addressed

Reviewer #2: All comments have been addressed

2. Is the manuscript technically sound, and do the data support the conclusions?

Reviewer #1: Yes

Reviewer #2: Yes

3. Has the statistical analysis been performed appropriately and rigorously? 

Reviewer #1: Yes

Reviewer #2: Yes

4. Have the authors made all data underlying the findings in their manuscript fully available?

Reviewer #1: Yes

Reviewer #2: Yes

5. Is the manuscript presented in an intelligible fashion and written in standard English?

Reviewer #1: Yes

Reviewer #2: Yes

6. Review Comments to the Author

Reviewer #1: (No Response)

Reviewer #2: The authors have addressed my comments in the revised manuscript, including the theoretical framework, methods and materials, results, discussion, and appendix. Particularly, they have thoroughly revised the entire discussion section and added sub-headings to enhance the manuscript's clarity, as I initially suggested.

Furthermore, the authors have made substantial improvements to both the structure and methodology of the manuscript based on the feedback received from the reviewers. In my opinion, the manuscript is now ready for acceptance. I hope that the authors will provide the required ethics statement and meet the Plos One journal's requirements to ensure that the manuscript can be published in the future.

7. PLOS authors have the option to publish the peer review history of their article (what does this mean?). If published, this will include your full peer review and any attached files.

Reviewer #1: **Yes: **Pham Duong Uyen Binh

Reviewer #2: No

---

## [Editor Report · Acceptance letter]

13 Oct 2023

PONE-D-23-12951R1 

Using the Theory of Planned Behavior to determine COVID-19 vaccination intentions and behavior among international and domestic college students in the United States 

Dear Dr. Liu:

I'm pleased to inform you that your manuscript has been deemed suitable for publication in PLOS ONE. Congratulations! Your manuscript is now with our production department. 

Kind regards, 

on behalf of

Prof. Dr. Khalid Mehmood 

Academic Editor

PLOS ONE